

# Exogenous application of 5-NGS increased osmotic stress resistance by improving leaf photosynthetic physiology and antioxidant capacity in maize

Deguang Yang[1], Zhifeng Gao[1], Yuqi Liu[1], Qiao Li[1], Jingjing Yang[1], Yanbo Wang[1], Meiyu Wang[1], Tenglong Xie[1], Meng Zhang[1] and Hao Sun[2]

[1] College of Agriculture, Northeast Agricultural University, Harbin, Heilongjiang, China
[2] College of Resources and Environment, Northeast Agricultural University, Harbin, Heilongjiang, China

Corresponding author
Hao Sun, sunhao@neau.edu.cn

## ABSTRACT

**Background**. Drought is a critical limiting factor affecting the growth and development of spring maize (*Zea mays* L.) seedlings in northeastern China. Sodium 5-nitroguaiacol (5-NGS) has been found to enhance plant cell metabolism and promote seedling growth, which may increase drought tolerance.

**Methods**. In the present study, we investigated the response of maize seedlings to foliar application of a 5-NGS solution under osmotic stress induced by polyethylene glycol (PEG-6000). Four treatment groups were established: foliar application of distilled water (CK), foliar application of 5-NGS (NS), osmotic stress + foliar application of distilled water (D), and osmotic stress + foliar application of 5-NGS (DN). Plant characteristics including growth and photosynthetic and antioxidant capacities under the four treatments were evaluated.

**Results**. The results showed that under osmotic stress, the growth of maize seedlings was inhibited, and both the photosynthetic and antioxidant capacities were weakened. Additionally, there were significant increases in the proline and soluble sugar contents and a decrease in seedling relative water content (RWC). However, applying 5-NGS alleviated the impact of osmotic stress on maize seedling growth parameters, particularly the belowground biomass, with a dry mass change of less than 5% and increased relative water content (RWC). Moreover, treatment with 5-NGS mitigated the inhibition of photosynthesis caused by osmotic stress by restoring the net photosynthetic rate (Pn) through an increase in chlorophyll content, photosynthetic electron transport, and intercellular $CO_2$ concentration (Ci). Furthermore, the activity of antioxidant enzymes in the aboveground parts recovered, resulting in an approximately 25% decrease in both malondialdehyde (MDA) and $H_2O_2$. Remarkably, the activity of enzymes in the underground parts exhibited more significant changes, with the contents of MDA and $H_2O_2$ decreasing by more than 50%. Finally, 5-NGS stimulated the dual roles of soluble sugars as osmoprotectants and energy sources for metabolism under osmotic stress, and the proline content increased by more than 30%. We found that 5-NGS played a role in the accumulation of photosynthates and the effective distribution of resources in maize seedlings.

**Conclusions**. Based on these results, we determined that foliar application of 5-NGS may improve osmotic stress tolerance in maize seedlings. This study serves as a valuable reference for increasing maize yield under drought conditions.

# INTRODUCTION

Maize (*Zea mays* L.) is an important crop worldwide that plays a vital role in food security and industrial applications (*Shiferaw et al., 2011*). Maize faces seasonally high temperatures and droughts, which have increased in frequency and severity under climate change and affect maize yield and quality (*Alkhalidi et al., 2023*). Northeastern China is an important grain-production base, with spring maize as the main crop. However, owing to factors such as climate change and geographic location, Heilongjiang Province often suffers from water shortages in spring, which affect maize seedlings during their crucial growth period (*Li et al., 2021*). Drought-induced osmotic stress can restrict the physiological activity of maize seedlings, posing challenges for proper growth during the seedling stage (*Moharramnejad et al., 2016*).

Drought-induced osmotic stress decreases the photosynthetic rate of maize seedlings, reducing the production of organic matter and leading to the accumulation of reactive oxygen species (ROS), which can cause oxidative damage to cells and lead to cell death (*Ghosh et al., 2021*). To alleviate the negative effects of drought, maize seedlings invest more material resources to synthesize the enzymes and osmotic adjustment substances needed by the antioxidant system, which inhibits vegetative growth (*Naeem et al., 2018*; *Song et al., 2019*). However, the exogenous application of silicon and polyamines, which have the potential to remove ROS, enhance the antioxidant system, reduce malondialdehyde (MDA) content, maintain membrane and unsaturated fatty acid integrity, and promote the accumulation of osmotic substances, has been shown to alleviate growth inhibition caused by drought (*Chang et al., 2022*; *Xu, Guo & Liu, 2022*). Furthermore, the exogenous application of coronatine and alginate oligosaccharides has been shown to increase drought tolerance in maize seedlings. These two exogenous substances alleviate the inhibition of photosynthesis and aboveground growth of maize caused by drought stress, thereby increasing the yield (*Guo et al., 2023*). In addition, through the exogenous spraying of urea on the leaves of maize seedlings under drought stress, changes in osmotic adjustment substances were induced, and the water content of the plant was maintained, showing drought resistance (*Gou et al., 2017*). Therefore, the application of exogenous substances can enhance the protective mechanisms of crop systems, promote the growth of maize during the seedling and later stages, and improve stress tolerance.

In particular, sodium 5-nitroguaiacol (5-NGS) is a cell-activating agent that regulates plant growth and has a strong osmotic effect (*Singh & Sharma, 1982*). It can quickly enter the plant body, promote the flow of plant protoplasts, accelerate plant rooting and germination, and promote growth, reproduction, and yield (*Al-Badawy et al., 1984*). Foliar application of 5-NGS was shown to have a positive effect on the nitrogen assimilation and metabolism of *Allium tuberosum* under light stress, and it improved the nutritional quality and promoted plant growth (*Li et al., 2014*). Additionally, common beans (*Phaseolus*

*vulgaris* L.) showed increased weight and yield when sprayed with preparations containing 5-NGS (*Kocira et al., 2017*). Studies have shown that 5-NGS is advantageous for regulating the quality, yield, and tolerance of some horticultural plants; however, it is not clear whether it can improve the yield, quality, and tolerance of field crops. As a $C_4$-type monocotyledonous field crop, maize utilizes its photosynthetic advantages, and nitrogen plays an important role in photosynthesis. Notably, 5-NGS is a nitrogen compound that drives protoplast flow and enhances cellular metabolism, with the potential to mobilize the photosynthetic advantages of maize under osmotic stress, thereby promoting the coordination of systems and enhancing drought tolerance (*Song et al., 2020*). Hence, there is a need to use maize to investigate the regulatory mechanisms of 5-NGS in field crops, particularly in relation to osmotic stress resistance.

In this study, a polyethylene glycol (PEG-6000) solution was used to simulate drought and induce osmotic stress, and 5-NGS was continuously sprayed onto the leaves of maize seedlings. The effects of foliar application of 5-NGS on the growth, photosynthetic, antioxidant, and osmoregulatory systems of maize seedlings were investigated within 7 d of the initiation of osmotic stress. We aimed to determine the effect of 5-NGS treatment on the growth and osmotic stress tolerance of maize seedlings.

# MATERIALS & METHODS

## Experimental design

This experiment was carried out in the maize cultivation physiology laboratory of the College of Agricultural, Northeast Agricultural University (Harbin, China, 126°73′E, 45°74′N). The maize cultivar Zhengdan 958, provided by the College of Agriculture, Northeast Agricultural University, was used as the test material. The maize seeds were disinfected with 1% NaClO for 10 min. After washing, seeds of the same size and fullness were selected and placed in a germination box with filter paper to avoid light germination, and the temperature was set to 28 °C. After germination, seeds with consistent growth were selected and sown in plastic basins (length: 50 cm; width: 38 cm; height: 20 cm). The basins were filled with vermiculite (depth: 12 cm), with eight seeds per basin. The basins were placed in a light incubator, with a light intensity of 400 $\mu$mol m$^{-2}$ s$^{-1}$, illumination for 16 h per day, a relative humidity of 70%, and a constant temperature of 25 °C. Four treatments were applied: (1) foliar application of distilled water (CK), (2) foliar application of 5-NGS (NS), (3) osmotic stress + foliar application of distilled water (D), and (4) osmotic stress + foliar application of 5-NGS (DN). There were five replications of each treatment, for a total of 20 basins and 160 seedlings. The seedlings were irrigated with 20% PEG-6000 (osmotic potential: $\Psi = -0.60$ MPa) to simulate osmotic stress treatment at the three-leaf stage (*Shereen et al., 2019*). The PEG-6000 solution was prepared using Hoagland nutrient solution. Normal water supply was provided using Hoagland complete nutrient solution (3.78 g $\cdot$kg$^{-1}$) (Table 1). The treatments were applied once every other day at a rate of 1 L per basin, for a total of two applications. Additionally, maize seedling leaves were sprayed with a 20 mg L$^{-1}$ solution of 5-NGS or distilled water at a volume of 80 mL per basin for a continuous period of 7 d, based on previously published application dosages (*Liu, 2012*).

**Table 1  Main nutritional components.**

| Inorganic salts | Concentration (mg L$^{-1}$) | Inorganic salts | Concentration (mg L$^{-1}$) |
|---|---|---|---|
| K$_2$SO$_4$ | 607 | Na$_2$B$_4$O$_7$ 10H$_2$O | 4.5 |
| NH$_4$H$_2$PO$_4$ | 115 | MnSO$_4$ | 2.13 |
| MgSO$_4$ | 493 | CuSO$_4$ | 0.05 |
| NaFe(EDTA) | 20 | ZnSO$_4$ | 0.22 |
| FeSO$_4$ | 2.86 | (NH$_4$)$_2$SO$_4$ | 0.02 |
| Ca(NO$_3$)$_2$ | 945 | | |

After the initiation of osmotic stress treatment, samples were collected at 3 d, 5 d, and 7 d to measure physiological parameters. Growth parameters were measured at 7 d. The samples were stored in a freezer at −80 °C for further analysis.

## Growth parameters

After 7 d of treatment, the plant height was measured as the length from the base of the maize seedlings to the top of the leaves using a ruler. After the freshly harvested corn roots were carefully cleaned with distilled water, three seedlings were taken from each treatment, and the root images were obtained using a Midcrystal i800 Plus root scanner. The root morphology, including the total root length, was analyzed using a Hangzhou Wanshen LA-S root analysis system, and the total leaf area of the whole maize seedling was measured using a YMJ-D leaf area measuring instrument (Zhejiang Topu Yunnong Technology Co., Ltd., Zhejiang, China). The shoots and roots of maize seedlings were inactivated at 105 °C for 30 min and then dried at 80 °C to a constant weight before being weighed.

We computed the relative effects of treatments by comparing the change in dry mass associated with the treatment to that of the control. The percent dry mass change (DMC) was calculated as follows:

$$DMC(\%) = 100 \times \frac{DM_{\text{treatment}} - DM_{\text{control}}}{DM_{\text{control}}}$$

where $DM_{\text{control}}$ and $DM_{\text{treatment}}$ are the dry biomasses of the control and the treatments, respectively.

## Relative water content

The relative water content (RWC) of the leaves was determined by weighing them. We collected a 0.5 g (fresh weight) sample of leaves and roots of different treatments and then saturated it through immersion in pure water for 24 h until swollen to determine the saturated weight. We then dried the sample at 80 °C for 24 h and weighed it again to determine the dry weight. The RWC was calculated using the following formula:

$$RWC(\%) = \frac{Fm - Dm}{Tm - Dm} \times 100,$$

where $Fm$, $Tm$, and $Dm$ are the fresh, saturated, and dry weights, respectively.

### Photosynthesis and gas exchange parameters

The net photosynthetic rate (Pn), intercellular $CO_2$ concentration (Ci), transpiration rate (Tr), and stomatal conductance (Gs) of the second expanded leaf of each plant were measured using a CI-340 handheld photosynthesis measurement system (CID, USA). The light intensity of the leaf chamber was 1400 $\mu$mol m$^{-2}$ s$^{-1}$, the temperature was 25 °C, and the humidity was 65%. After 20 min of dark treatment, chlorophyll fluorescence parameters were measured using a plant light efficiency analyzer (Pocket Pea, Hansatech, UK). The soil plant analysis development (SPAD) value, which is a measurement of leaf chlorophyll concentration, was determined using a MultispeQ V2.0 device (Beijing Huinuored Technology Co., Ltd., Beijing, China).

### Osmotic adjustment substance content

The proline content in the maize leaves and roots was determined using the sulfosalicylic acid method, as previously described by *Bates, Ra & Teare (1973)*. The soluble protein content in maize leaves and roots was determined using the Coomassie brilliant blue G-250 method (*Azevedo Neto, Prisco & Gomes-Filho, 2009*). The soluble sugar content in the maize leaves and roots was determined using anthrone colorimetry (*Gurrieri et al., 2020*).

### MDA and $H_2O_2$ contents

The MDA content in maize leaves and roots was measured using the thiobarbituric acid method, which evaluates membrane lipid peroxidation as an indicator of damage by ROS (*Mohammadkhani & Heidari, 2007*). The absorbance was then measured at 532 and 600 nm. The $H_2O_2$ content was determined according to the method of *Alexieva et al. (2001)*. Notably, MDA and $H_2O_2$ are well-known indicators of oxidative stress, such as that occurring during osmotic stress.

### DAB histochemical detection

Staining of $H_2O_2$ in maize seedling leaves by diaminobenzidine(DAB), it was performed by immersing the sheared leaves in 1.5 mg ml$^{-1}$ DAB solution at pH 3.8, evacuating them for 30 min, and infiltrated for 12 h under darkness. The leaves were degreened by boiling using anhydrous ethanol and photographed.

### Antioxidant enzyme activity

The activities of superoxide dismutase (SOD), peroxidase (POD), and catalase (CAT) in maize leaves and roots were measured using commercial kits (Beijing Solarbio Science & Technology Co., Ltd., China) according to the manufacturer's instructions. The activities of SOD, POD, and CAT were recorded at wavelengths of 560, 470, and 240 nm, respectively.

### Data analysis

All data were analyzed through one-way analysis of variance using SPSS 25.0 software. The significance of differences between treatments was determined using Duncan's test ($P < 0.05$). All charts in this study were plotted using Origin 2021 software.

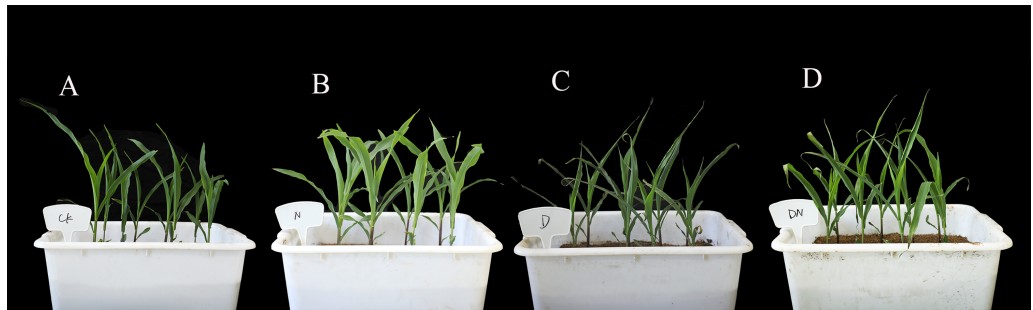

**Figure 1** **Photographs of maize seedling growth in different treatments.** A, B, C, and D represent maize seedlings under four treatments: CK, NS, D, and DN (at 7 days of treatment; leaf age: 3 leaves - 4 leaves), respectively.

## RESULTS

### Maize seedling growth

Figure 1 shows the growth of maize seedlings (treatment 7d) in each treatments. Compared to CK, treatment NS did not significantly affect the growth parameters. However, treatment D significantly decreased the growth index, with plant height and root length decreasing by 34.21% and 26.51% (Figs. 2C and 2E), respectively. The leaf area under treatment D also decreased by 59.51% (Fig. 2D) compared to that under CK. Additionally, the accumulation of aboveground and belowground dry masses under treatment D decreased by 43.20% and 32.07%, respectively (Fig. 2A), compared to those under CK. However, maize seedlings exhibited better growth under treatment DN than under treatment D, as evidenced by significant increases in plant height and root length of 13.34% and 12.65% (Figs. 2C and 2E), respectively. The leaf area under treatment DN also experienced a significant increase of 33.03% (Fig. 2D), whereas the aboveground and belowground dry mass accumulation had significant increases of 24.90% and 31.06%, respectively (Fig. 2A), compared to those measurements under treatment D. Notably, the biomass was restored to control levels under treatment DN, with the change in belowground biomass not exceeding 5% compared with that under the control (Fig. 2B). Furthermore, under treatment D, the RWC in the leaves and roots decreased significantly compared to that under CK. Meanwhile, under treatment DN, the RWC in the leaves and roots significantly increased by 7.34% and 7.26%, respectively, compared to that under treatment D (Fig. 3).

### Leaf photosynthesis

At 3 d, the SPAD values under treatment D exhibited a decreasing trend compared to those under CK, with a significant reduction of 21.37% observed at 7 d. Conversely, the SPAD values under treatment DN showed significant increases of 11.16%, 19.23%, and 13.72% at 3 d, 5 d, and 7 d, respectively, compared to those under treatment D (Fig. 4A). Furthermore, at 3 d, the Pn, Tr, Ci, and Gs values under treatments D and DN showed significant decreases of more than 52% compared to those under CK. At 5 d, the Pn, Tr, Ci, and Gs values under treatment D showed significant decreases of more than 54% compared to those under CK, whereas these values showed significant increases of less than

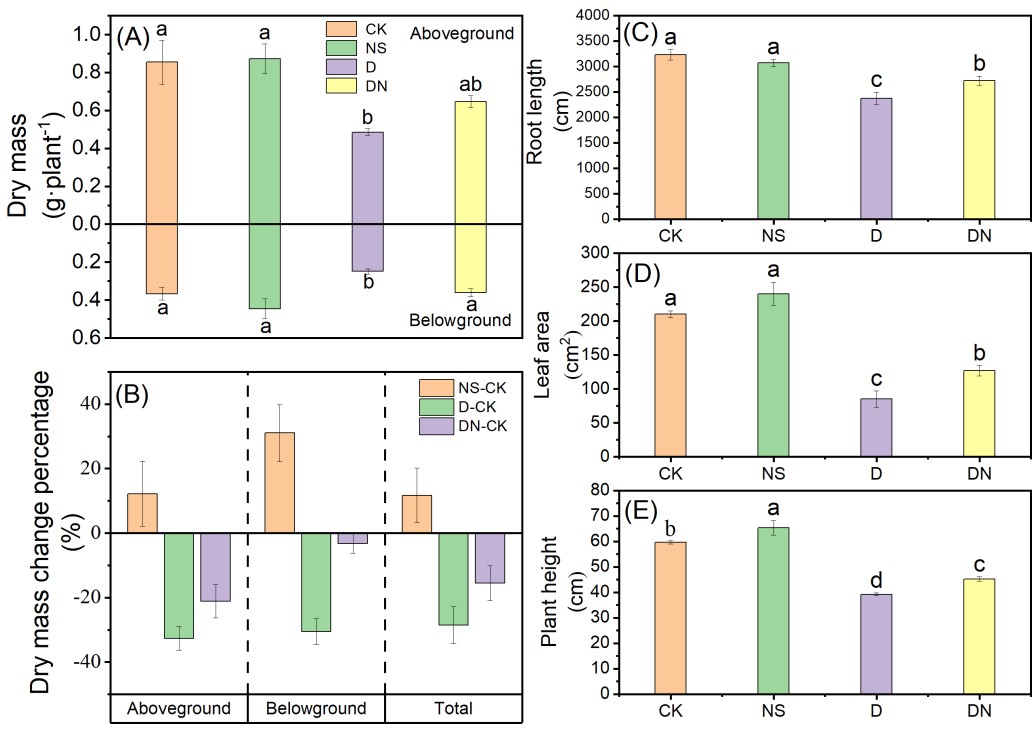

**Figure 2 Growth parameters of maize seedlings under different treatments.** Effect of dry mass (A), dry mass change (percentage, relative to control) (B), root length (C), leaf area (D) and plant height (E) of maize seedlings. Treatment labels are NS-CK, D-CK and DN-CK represent the dry mass changes between the corresponding treatments and the control, respectively. Treatment labels are CK: no stress; NS: no stress and spraying 5-NGS; D:osmotic stress; DN:osmotic stress and spraying 5-NGS. There was no significant difference in the mean of the same letters when $P \geq 0.05$. (Figs. 3, 4, 5, 6 and 8 same).

48% under treatment DN. Finally, at 7 d, the Pn, Tr, Ci, and Gs values under treatment D showed significant decreases of more than 65% compared to those under CK, whereas these values showed significant increases of more than 55.20% under treatment DN compared to those under treatment D (Figs. 4C, 4D, 4E, and 4F). Additionally, it was observed that compared to those under CK, the $F_v/F_m$ values of maize seedlings decreased by 3.24% and 6.4% under treatment D at 5 d and 7 d, respectively. However, under treatment DN, these values increased significantly by 2.88%, 4.32%, and 5.96% at 3 d, 5 d, and 7 d, respectively (Fig. 4B), compared to those under treatment D.

## Osmotic adjustment of maize seedlings

The proline content in the leaves significantly increased under treatment D, with increases of 67.57%, 72.12%, and 56.16% at 3 d, 5 d, and 7 d, respectively (Fig. 5A), compared to those under CK. Moreover, treatment DN further elevated the proline content in the leaves, with significant increases of 10.58%, 17.97%, and 33.90% at 3 d, 5 d, and 7 d, respectively (Fig. 5A), compared to those under treatment D. In the roots, the proline content under treatment D decreased by 6.33%, 29.17%, and 7.28% at 3 d, 5 d, and 7 d, respectively (Fig. 5A), compared to that under CK. Conversely, treatment DN resulted in significant

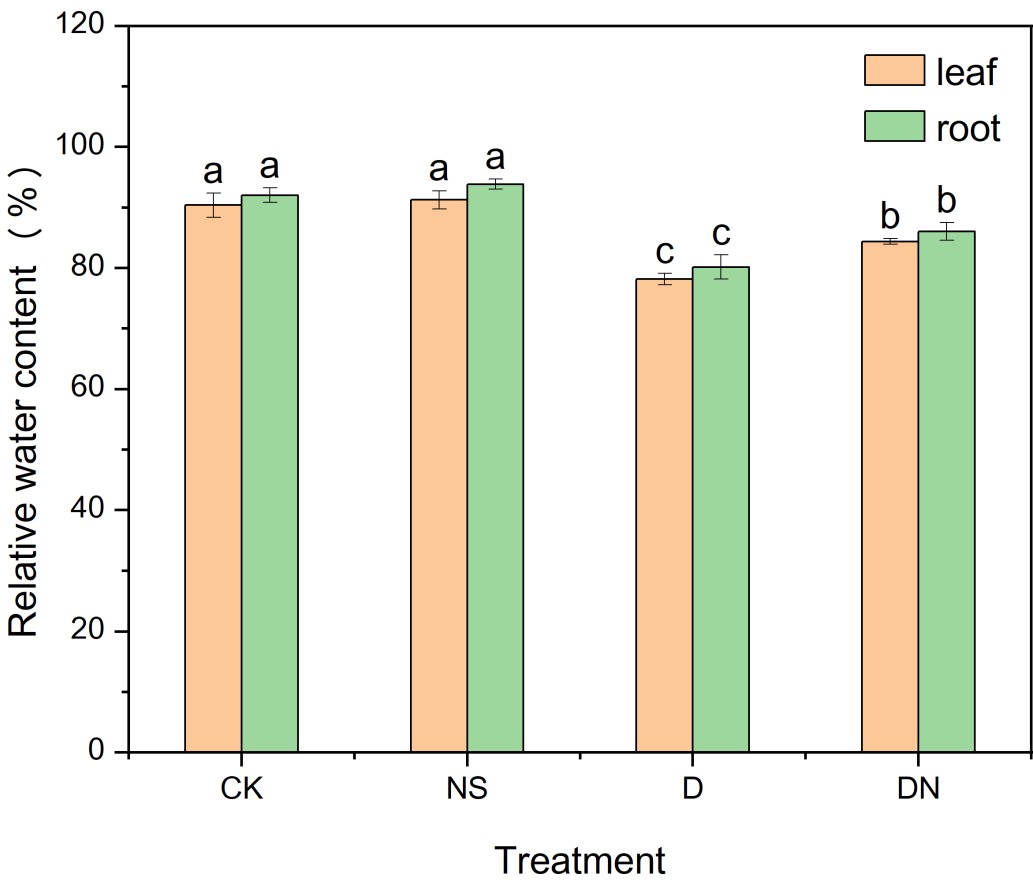

**Figure 3   Relative water content of maize seedlings under different treatments.** Effect of relative water content (RWC) of maize seedlings.

increases in the proline content in the roots of 15.33%, 30.04%, and 34.17% at 3 d, 5 d, and 7 d, respectively (Fig. 5A), compared to those under treatment D.

The soluble protein content in the leaves significantly decreased under treatment D, with reductions of 11.59%, 12.67%, and 16.52% at 3 d, 5 d, and 7 d, respectively (Fig. 5B), compared to that under CK. In contrast, treatment DN resulted in a significant increase in the soluble protein content in the leaves, with increases of 9.55%, 13.10%, and 4.77% at 3 d, 5 d, and 7 d, respectively (Fig. 5B), compared to that under treatment D. However, in the roots, exogenous spraying with 5-NGS did not have a significant effect on the soluble protein content (Fig. 5B).

Compared to CK, treatment NS significantly increased the soluble sugar content in both the leaves and roots. Similarly, treatment D also resulted in an increase in soluble sugar content compared to that under CK. Specifically, the soluble sugar content in the leaves showed a significant increase of 30.15%, 32.95%, and 30.31% at 3 d, 5 d, and 7 d, respectively, whereas in roots, the increase was 11.64%, 31.24%, and 15.59% (Fig. 5C). In contrast, compared to that under treatment D, the soluble sugar content in the leaves under treatment DN exhibited a decrease of 7.99%, 25.68%, and 17.95% at 3 d, 5 d, and

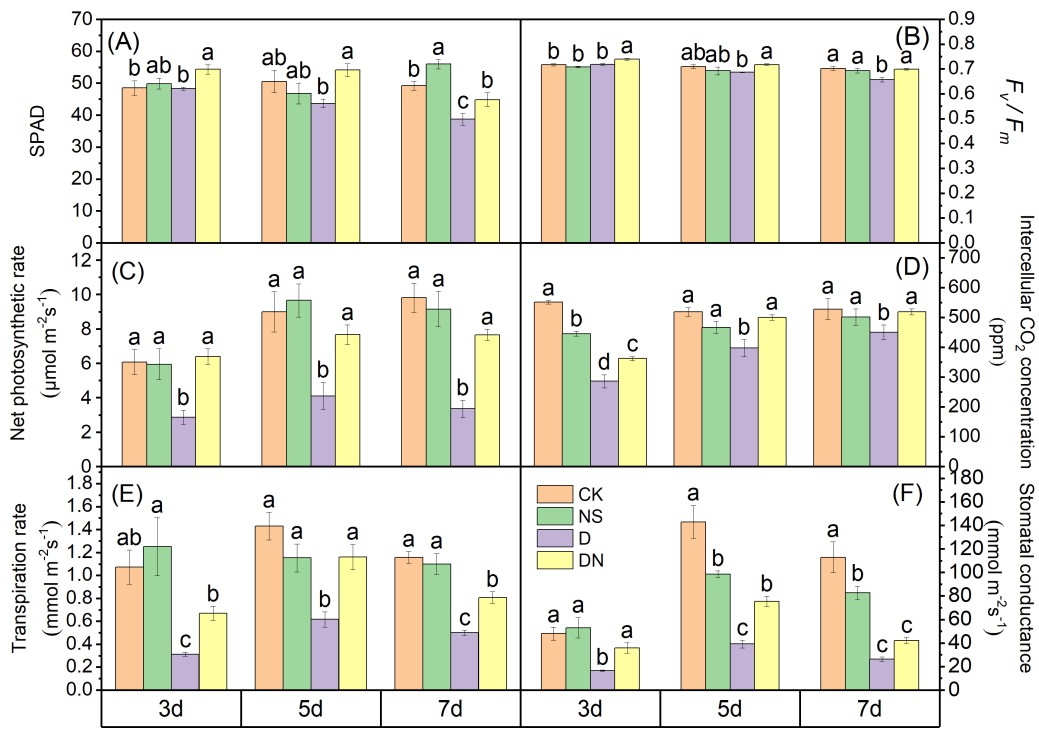

**Figure 4** **Photosynthetic capacity of maize seedlings under different treatments.** Effect of SPAD (A), *Fv/Fm* (B), net photosynthetic rate (C), intercellular $CO_2$ concentration (D), transpiration rate (E), stomatal conductance (F) of maize seedlings.

7 d, respectively, with the difference being significant at 5 d and 7 d (Fig. 5C). Additionally, the soluble sugar content in the roots showed a significant decrease of 12.77%, 21.73%, and 31.74% at 3 d, 5 d, and 7 d, respectively (Fig. 5C).

## Oxidative stress of maize seedlings

Under treatment NS, the MDA content in the leaves was significantly reduced by 18.37% and 26.85% at 5 d and 7 d, respectively, compared to that under CK (Fig. 6A). Conversely, under treatment D, the MDA content at 5 d and 7 d increased by 14.04% and 20.90%, respectively, compared to that under CK. Additionally, the $H_2O_2$ content under treatment D increased significantly by 31.88%, 38.79%, and 16.68% at 3 d, 5 d, and 7 d, respectively, compared to that under CK (Fig. 6B). Furthermore, under treatment DN, the MDA content decreased by 3.70%, 20.89%, and 26.85% at 3 d, 5 d, and 7 d, respectively, compared to that under treatment D, with the difference being significant at 5 d and 7 d (Fig. 6A). Moreover, under treatment DN, the $H_2O_2$ content decreased by 18.92%, 38.54%, and 23.31%, respectively (Fig. 6B), compared to that under treatment D. In the roots, the MDA content increased significantly under treatment D compared to that under CK, and the $H_2O_2$ content increased significantly by 57.66%, 43.45%, and 58.71% at 3 d, 5 d, and 7 d, respectively (Fig. 6B). Compared with that under treatment D, the MDA content under treatment DN decreased significantly by 55.56%, 26.60%, and 70.42% at 3 d, 5 d, and 7 d, respectively, and the $H_2O_2$ content decreased significantly by 53.74%, 37.99%, and 40.69%,

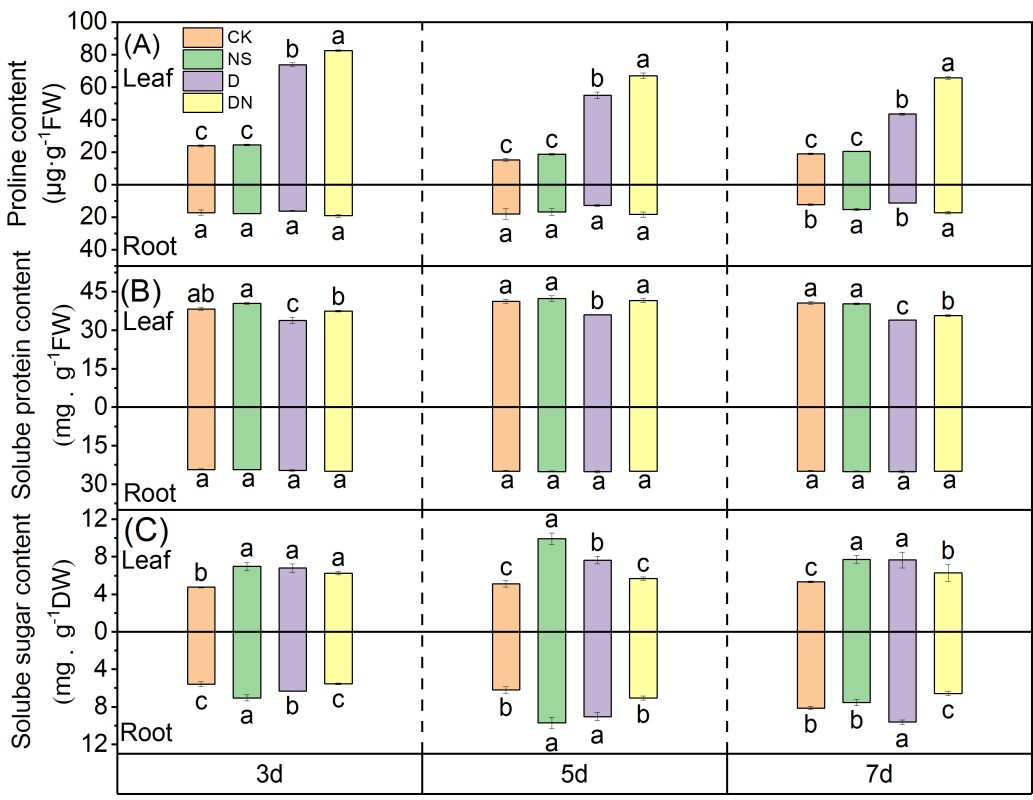

**Figure 5 Osmoregulatory substances in maize seedlings under different treatments.** Effect of proline content (A), soluble protein content (B), soluble sugar content (C) in leaf and root of maize seedlings.

respectively (Figs. 6A and 6B). The *in situ* detection of $H_2O_2$ *in vivo* histochemistry by DAB staining experiments showed that leaves under treatment CK and treatment NS contained very little $H_2O_2$ and light brown color, while leaves under treatment D contained the most $H_2O_2$ and the darkest brown color, which gradually deepened with time. Compared with treatment D, leaves under treatment DN had significantly less brown color and decreased $H_2O_2$ content (Fig. 7).

## Antioxidant activities of maize seedlings

Under treatment D, the activity of POD in the leaves showed a time-dependent decrease, with reductions of 37.57%, 57.07%, and 88.66% at 3 d, 5 d, and 7 d, respectively, compared to that under CK. The differences between the treatments were significant (Fig. 8A). However, compared to the POD activity in the leaves under treatment D, that under treatment DN was significantly increased by 16.89%, 53.50%, and 77.32% at 3 d, 5 d, and 7 d, respectively (Fig. 8A). The POD activity in the roots under treatment D was higher than that under CK, and the difference was significant at 3 d and 5 d of treatment (Fig. 8A). However, compared to the POD activity in the roots under treatment D, that under treatment DN showed significant increases of 13.90%, 39.56%, and 34.21% at 3 d, 5 d, and 7 d, respectively.

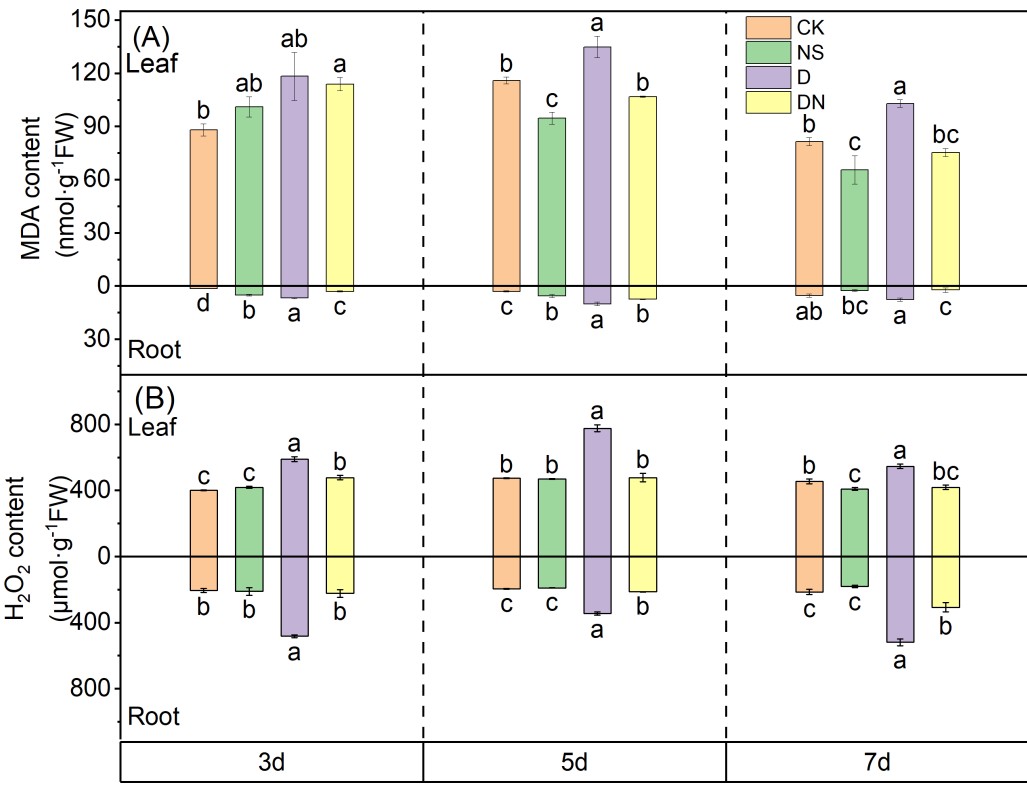

**Figure 6  Oxidative stress in maize seedlings under different treatments.** Effect of MDA content (A), H₂O₂ content (B) in leaf and root of maize seedlings.

Compared to that under CK, the SOD activity in the leaves of maize seedlings under treatment NS was significantly increased by 19.63% and 27.21% at 3 d and 7 d, respectively. Conversely, under treatment D, SOD activity in the leaves showed an upward trend with increasing treatment time. Compared to that under CK, the SOD activity in the leaves under treatment D was significantly increased by 27.37%, 48.11%, and 57.99% at 3 d, 5 d, and 7 d, respectively (Fig. 8B). Furthermore, compared to that under treatment D, the SOD activity in the leaves under treatment DN significantly increased by 21.78% at 5 d (Fig. 8B). In the roots, there was no significant difference in the SOD activity between the NS treatment and CK. Additionally, compared to that under treatment D, the SOD activity in the roots increased under treatment DN, but the difference was not significant (Fig. 8B).

Under treatment D, the CAT activity in the leaves decreased significantly by 63.60%, 67.49%, and 71.73% at 3 d, 5 d, and 7 d, respectively, compared to that under CK. The differences between the treatments were significant (Fig. 8C). Conversely, compared with that under treatment D, the CAT activity in the leaves increased significantly under treatment DN. Specifically, at 5 d and 7 d, the CAT activity was significantly increased by 57.42% and 57.43%, respectively. In the roots, treatment NS had a significant effect on CAT activity compared to that of CK at 5 d and 7 d (Fig. 8C). Additionally, under treatment D, CAT activity in the root system showed an upward trend with increasing treatment

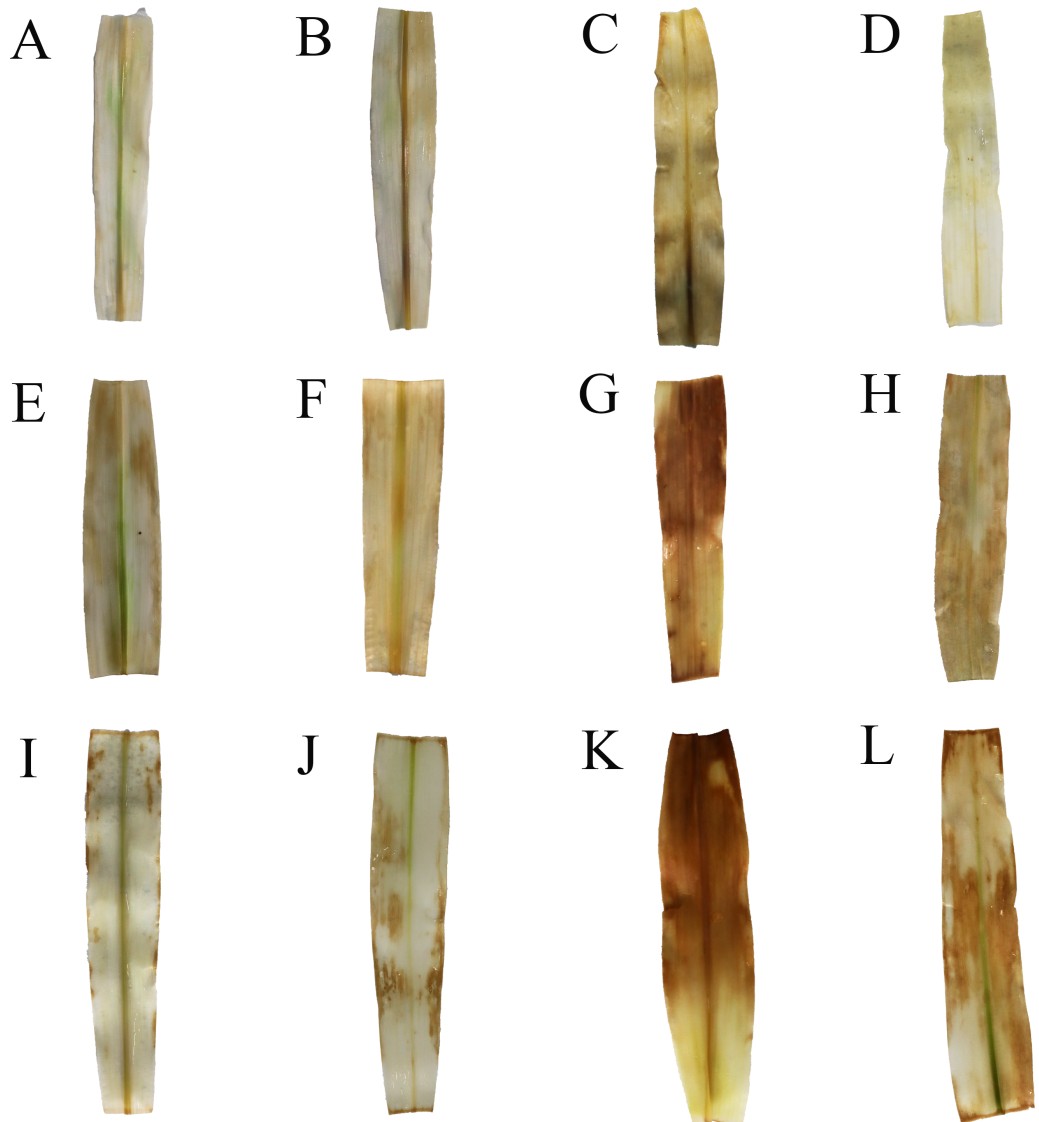

**Figure 7 DAB tissue staining.** The $H_2O_2$ content in leaf tissues. A, B, C, D indicate CK, NS, D, DN treatments at 3d; E, F, G, H indicate CK, NS, D, DN treatments at 5d; I, J, K, L indicate CK, NS, D, DN treatments at 7d. The higher the $H_2O_2$ content, the darker the brown color.

time, but it was consistently lower than that under CK (Fig. 8C). However, compared with treatment D, treatment DN significantly increased CAT activity in the roots. Notably, there were increases of 43.64%, 69.29%, and 53.12% after 3 d, 5 d, and 7 d of treatment, respectively (Fig. 8C).

## Principal component analysis

Principal component analysis (PCA) of the leaf metrics revealed that PC1 and PC2 accounted for 68.4% and 19.3% of the variance, respectively, resulting in a combined variance of 87.7% (Fig. 9). This finding indicates that the first two principal components

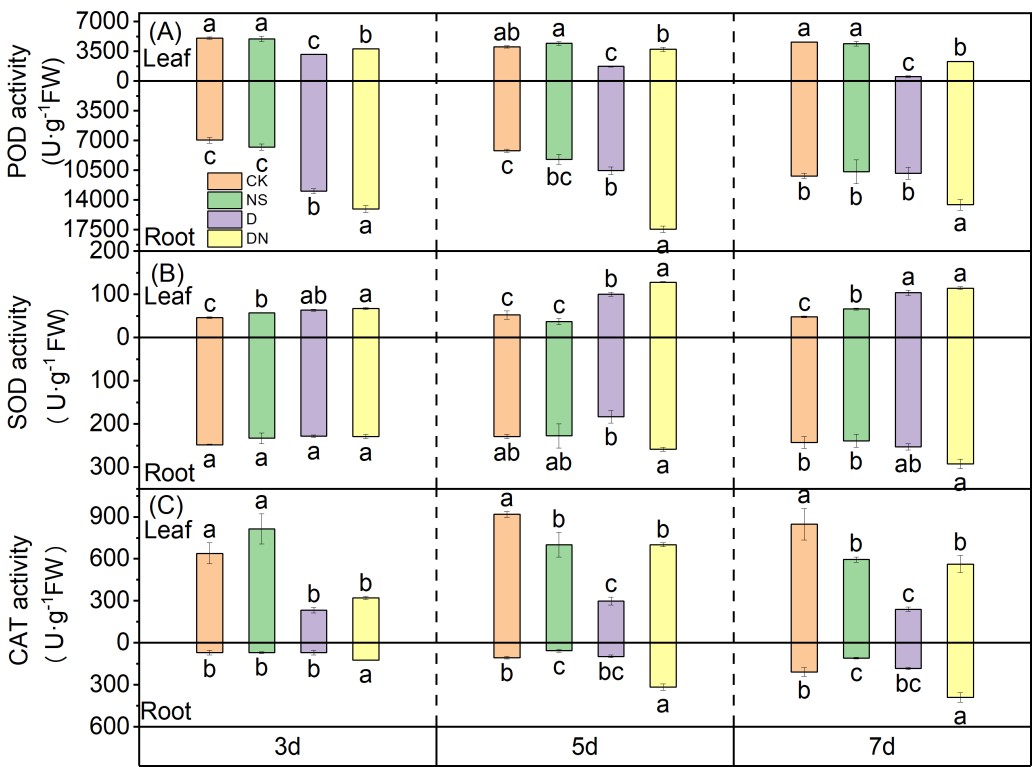

**Figure 8  Antioxidant enzyme activities in maize seedlings under different treatments.** Effect of POD activity (A), SOD activity (B), CAT activity (C) in leaf and root of maize seedlings.

effectively captured the variability observed in each physiological index. Notably, variables related to antioxidant enzyme activity, photosynthetic indices, and osmoregulatory substances exhibited strong positive loadings on PC1. This suggests that these factors play a crucial role in maize seedling growth. Similarly, in the case of roots, PC1 and PC2 accounted for 49.7% and 39.1% of the variation, respectively, resulting in a combined variance of 86.8% (Fig. 10). The variables associated with antioxidant enzyme activity, photosynthetic indices, and osmoregulatory substances had even higher positive loadings for both PC1 and PC2, emphasizing their significance as indicators in this context.

## DISCUSSION

Drought is widely recognized as a factor that hinders the growth of maize seedlings and disrupts their developmental processes, including photosynthesis and metabolism. This disruption affects plant biomass and morphological plasticity, which are key indicators of drought stress (*Dos Santos et al., 2022*; *Kocira et al., 2017*). Previous studies on maize have demonstrated a significant reduction in morphological plasticity, including that of plant height, number of branches, and leaf area, under osmotic stress (*Saad-Allah et al., 2021*; *Seleiman et al., 2021*). Our findings showed a substantial reduction in leaf area under osmotic stress conditions. Maize seedlings exposed to osmotic stress exhibited limited growth, characterized by reduced RWC, reduced plant height, shortened root length, and

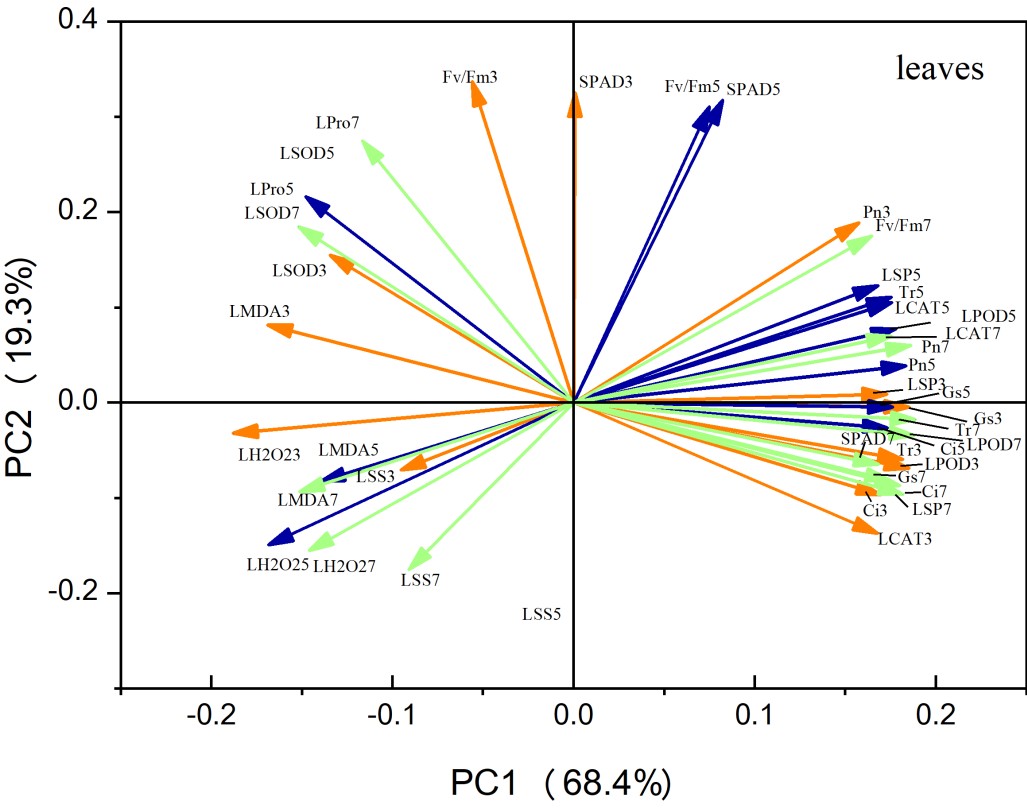

**Figure 9** **Principal component analysis of leaf indicators.** The orange line represents the treatment on the third day, the blue line represents the treatment on the fifth day, and the green line represents the treatment on the seventh day (Fig. 10 same).

decreased dry matter accumulation (Figs. 2 and 3), which is consistent with earlier findings (*Zhao et al., 2018*). This stress response resulted in constrained material accumulation and inhibited root growth. However, following the exogenous application of 5-NGS to maize seedling leaves under osmotic stress, growth significantly improved, with only minor changes observed in belowground biomass compared to that under the control (Fig. 2B), and the leaf and root RWC increased (Fig. 3). Although the application of 5-NGS alleviated reductions in morphological plasticity, it did not fully restore plasticity to the control levels. However, biomass levels were restored to control levels, with the change rate of belowground biomass under 5-NGS treatment not exceeding 5% of that under the control (Fig. 2B). This indicates that it is favorable for organic matter to be stored in the root organs to meet the root material demand under osmotic stress. Therefore, we posit that 5-NGS contributes to the recovery of photosynthesis, synthesis of osmotic substances and enzymes, substance accumulation, and rational distribution of resources in maize seedlings. These physiological responses ultimately provide feedback for belowground growth, ensuring root development, improved nutrient absorption, and increased tolerance to osmotic stress.

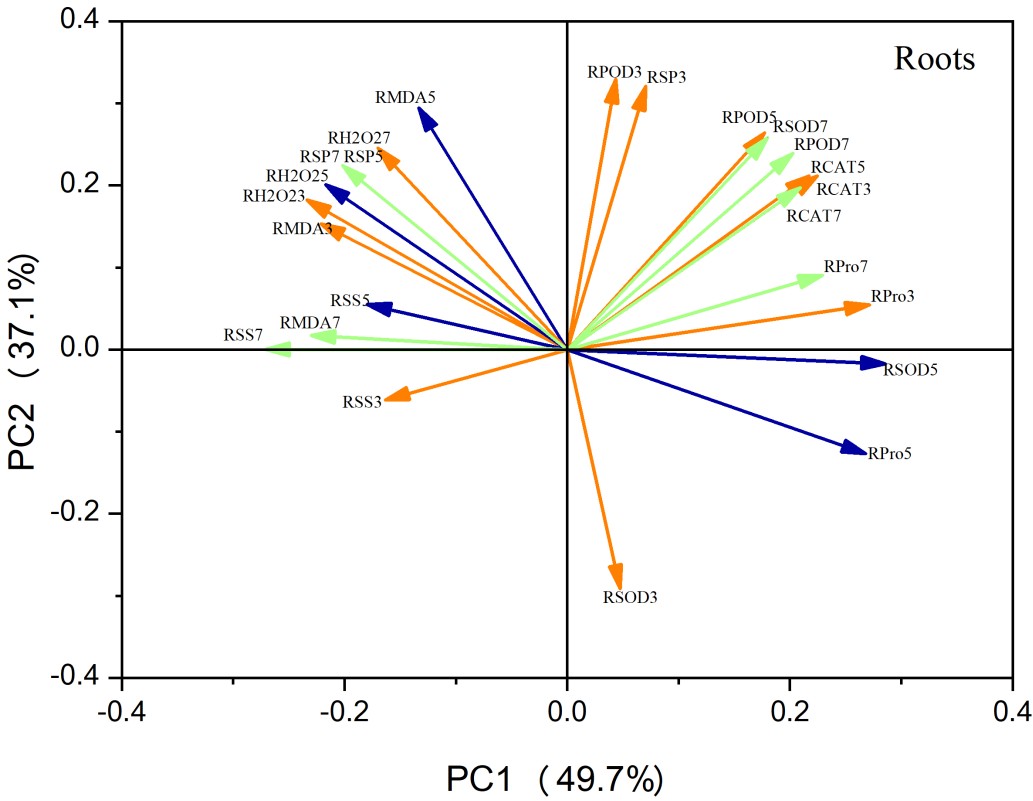

**Figure 10 Principal component analysis of root indicators.**

## 5-NGS alleviates the physiological inhibitory effect of osmotic stress on aboveground growth of maize seedlings

Drought impedes stomatal function and photosynthetic capacity, leading to growth limitations (*Li et al., 2020*). Stomatal limitation is a crucial factor hindering $CO_2$ diffusion from the air to the mesophyll under drought stress, as evidenced by reduced Pn, Gs, Ci, and Tr (*Naseer et al., 2022*). Our results showed that osmotic stress significantly suppressed Gs, limited Ci, and significantly reduced Tr and Pn. However, Pn, Gs, Ci, and Tr were restored after the application of 5-NGS, suggesting an ability to alleviate the stomatal limitation of maize leaves caused by osmotic stress (Fig. 4). Therefore, the recovery of Pn can be attributed to the regulation of Gs by 5-NGS, which maintains the supply of $CO_2$ and replenishes the raw material for photosynthesis. The inhibition of photosynthetic capacity by drought stress is also manifested in damage to the photosystems and degradation of photosynthetic pigments (*Jin et al., 2023*). Our results showed that SPAD levels were reduced under the osmotic stress treatments compared to those in the control, and $F_v/F_m$ decreased, resulting in the inhibition of PSII (Figs. 4A and 4B). In contrast, the increase in $F_v/F_m$ under the 5-NGS treatments indicated that it promoted PSII light energy conversion efficiency, increased reaction center activity, and restored Pn (Fig. 4B). Maize seedlings absorb 5-NGS and synthesize chlorophyll, which may activate the light-harvesting mechanism of PSII in the primary reactions of photosynthesis, thus enhancing the efficiency of electron

transport (*Batool et al., 2022*). In addition, nitrophenol compounds are known to enhance crop photosynthesis (*Kazda et al., 2014*). Consequently, it may be that 5-NGS restores the photosynthetic capacity of maize seedlings by mobilizing the light- and dark-response phases under osmotic stress to establish a material basis for the resistance system and enhance resistance to osmotic stress (*Kazda et al., 2014*). This defensive strategy reflects an adaptation to arid environments.

Proline reduces oxidative stress and stimulates antioxidant enzyme activity by chelating reactive oxygen radicals. Additionally, it forms hydrophobic membranes of protein molecules to protect their structure, maintains proteolysis, and plays an osmoregulatory role in plant cells under adverse conditions (*Yang et al., 2021*). Meanwhile, the soluble sugar content not only enhances the osmotic adjustment capability but also provides essential carbon reserves for plants (*Ergo et al., 2021*). Moreover, drought-stressed crops upregulate the expression of sugar transporter proteins to promote sugar transport and meet organ energy requirements (*Kaur et al., 2021*). In our study, the proline and soluble sugar contents in maize seedling leaves significantly increased under osmotic stress (Figs. 5A and 5C), highlighting an important strategy for maintaining leaf osmotic pressure and ensuring normal physiological processes in the leaves (*Du et al., 2020*). Conversely, a substantial decrease in soluble proteins during drought may be attributed to protein decomposition, synthesis inhibition, or conversion to proline in the leaves. Foliar application of 5-NGS to maize seedlings under osmotic stress conditions led to increases in the proline and soluble protein contents in leaves compared to those in untreated leaves, whereas the sugar content decreased to control levels (Figs. 5A, 5B, and 5C). Therefore, it may be that the application of 5-NGS drives sugar translocation to provide energy for replenishment of proline, which maintains proteolysis and ensures that the osmotic agent accumulates sufficiently in the maize seedling to maintain cellular tension (*Rosinger, Wilson & Kerr, 1984*). In addition, the large accumulation of proline acts as a protein- and enzyme-dehydrating agent and as a protective agent for various enzymes, restoring protein and enzyme activities (*Paleg, Stewart & Bradbeer, 1984*; *Silva-Ortega et al., 2008*).

Activating the antioxidant-protective enzyme system is a critical strategy for plants to sustain normal photosynthetic physiological activities and self-protection (*Logan et al., 2006*). The MDA and $H_2O_2$ content are commonly used to measure the degree of damage to membrane structure and function in maize (*Singh et al., 2022*). In our study, under osmotic stress, the MDA and $H_2O_2$ contents in leaves increased significantly, whereas the activities of POD and CAT decreased with prolonged osmotic stress time, and SOD activity increased (Figs. 6A and 6B; 8A, 8B, and 8C). However, our results revealed that the application of 5-NGS to maize seedlings under osmotic stress resulted in the restoration of POD and CAT activities over time, approaching the control levels. Additionally, SOD activity further increased, exceeding 30% under osmotic stress (Figs. 8A, 8B, and 8C). The MDA and $H_2O_2$ contents in maize seedling leaves under osmotic stress decreased to control levels after prolonged exposure to 5-NGS (Figs. 6A and 6B). These results suggest that under the stimulation of 5-NGS, maize seedlings integrate and utilize the stored energy and material resources for photosynthesis (*Li et al., 2015*).

## 5-NGS improves the belowground growth of maize seedlings under osmotic stress

Plant roots, which are critical for water and mineral nutrient absorption, play indispensable roles in plant growth and development (*Hodge et al., 2009*). Levels of osmoregulatory substances in roots are indicative of a plant's drought tolerance (*Ozturk et al., 2021*). In our study, under osmotic stress, the proline and soluble protein contents of maize seedling roots remained relatively unchanged, whereas the soluble sugar content significantly increased compared to that under the control (Figs. 5A, 5B, and 5C). Thus, soluble sugars have emerged as primary osmotic regulators in maize seedling roots under osmotic stress. However, after the application of 5-NGS under osmotic stress conditions, the proline content in maize seedling roots significantly increased over time, surpassing that of the control group (Fig. 5A). Simultaneously, the soluble sugar content decreased under the influence of 5-NGS compared to that without treatment, returning to control levels. This shift may be attributed to the 5-NGS activating photosynthesis, maintaining soluble sugar levels, and mobilizing the remaining soluble sugars for proline production (*Joyce, Aspinall & Paleg, 1992*; *Moustakas et al., 2011*).

Root vigor and protective enzyme activity significantly affect the growth and physiological functions of seedlings (*Dawood et al., 2022*). Roots are the first organs to perceive soil drought, and any negative responses in the roots affect the growth and function of aboveground organs, ultimately restricting overall plant development (*Li et al., 2021*). Our results align with this observation, as the MDA and $H_2O_2$ contents in maize seedling leaves were only 16–30% and 20–60% higher, respectively, under osmotic stress than under control conditions, whereas these contents in the roots of stressed plants exceeded those in the unstressed plants by 200–400% and 140%, respectively (Figs. 6A and 6B). Additionally, protective enzyme activity in the aboveground and belowground parts varied significantly under osmotic stress conditions, with SOD showing strong activity in the aboveground parts and POD showing strong activity in the roots (Figs. 8A and 8B). Following the application of 5-NGS to maize seedlings under osmotic stress conditions, the MDA and $H_2O_2$ contents in the roots significantly decreased compared to those without treatment (Figs. 6A and 6B). The promotional effect of 5-NGS on the activity of maize seedling protective enzymes was more pronounced in the roots than in the leaves (Fig. 8). In particular, SOD, POD, and CAT activities in maize seedling roots significantly exceeded the control levels under osmotic stress with prolonged exposure to 5-NGS, whereas in the leaves, POD and CAT activities did not fully recover to control levels (Fig. 8). This suggests that the root system is the primary organ protected from osmotic stress. When exogenous carbohydrates are present, crops absorb, utilize, store, or metabolize them, simultaneously upregulating the activity of protective enzymes (*Couée et al., 2006*; *Zhang et al., 2021*). Therefore, under adverse conditions, maize seedlings allocate most of their stored resources to root growth and protective enzyme synthesis through the mobilization of 5-NGS metabolism. Furthermore, 5-NGS improves substance accumulation, ultimately benefiting belowground growth and enhancing osmotic stress tolerance.

## CONCLUSIONS

We assert that 5-NGS promotes the synthesis of osmotic substances, activates antioxidant enzyme activity, improves antioxidant capacity, reduces the accumulation of ROS and other harmful substances, and coordinates the rational allocation of resources. These physiological responses ultimately support belowground growth, ensuring root development and improving osmotic stress tolerance.

## ACKNOWLEDGEMENTS

Thanks to all the staff of the Maize Adversity Physiology Group of Northeast Agricultural University.

### Funding

This work was supported by the National Natural Science Foundation of China under Grant. (32301939), the Heilongjiang Provincial Natural Science Foundation under Grant (LH2021C024) and the Northeast Agricultural University under Grant (20QC02). The funders had no role in study design, data collection and analysis, decision to publish, or preparation of the manuscript.

### Grant Disclosures

The following grant information was disclosed by the authors:
The National Natural Science Foundation of China: 32301939.
The Heilongjiang Provincial Natural Science Foundation: LH2021C024.
The Northeast Agricultural University under Grant: 20QC02.

### Competing Interests

The authors declare there are no competing interests.

### Author Contributions

- Deguang Yang conceived and designed the experiments, performed the experiments, analyzed the data, prepared figures and/or tables, authored or reviewed drafts of the article, and approved the final draft.
- Zhifeng Gao conceived and designed the experiments, performed the experiments, analyzed the data, prepared figures and/or tables, authored or reviewed drafts of the article, and approved the final draft.
- Yuqi Liu performed the experiments, prepared figures and/or tables, and approved the final draft.
- Qiao Li analyzed the data, prepared figures and/or tables, and approved the final draft.
- Jingjing Yang performed the experiments, analyzed the data, prepared figures and/or tables, and approved the final draft.
- Yanbo Wang performed the experiments, analyzed the data, prepared figures and/or tables, and approved the final draft.

- Meiyu Wang analyzed the data, prepared figures and/or tables, and approved the final draft.
- Tenglong Xie analyzed the data, authored or reviewed drafts of the article, support for funding, and approved the final draft.
- Meng Zhang analyzed the data, prepared figures and/or tables, and approved the final draft.
- Hao Sun performed the experiments, authored or reviewed drafts of the article, and approved the final draft.

## Data Availability

The raw measurements are available in the Supplementary File.

## Supplemental Information

Supplemental information for this article can be found online at http://dx.doi.org/10.7717/peerj.17474#supplemental-information.

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
