# Peer review of "Exogenous application of 5-NGS increased osmotic stress resistance by improving leaf photosynthetic physiology and antioxidant capacity in maize"

_PeerJ, doi:10.7717/peerj.17474_

## Round 0.1 · original submission · Major Revisions

Dear Dr. Gao,
Thank you for your submission to PeerJ.
It is my opinion as the Academic Editor for your article - Exogenous application of 5-NGS increased osmotic stress resistance by improving leaf photosynthetic physiology and antioxidant capacity in maize - that it requires a number of Major Revisions.

Reviewer 1 ·

Basic reporting

Dear Authors,
As I have gone through the manuscript entitled, "Exogenous application of 5-NGS increased osmotic stress resistance by improving leaf photosynthetic physiology and antioxidant capacity in maize"
I would like to conclude now that the language style used for manuscript writing is not fit for research publication. The current language style is extremely poor in several places to understand the messages. In the current form, it may not be acceptable in Peer J for publication. I suggest authors take the help of a language expert to fully revise the manuscript and resubmit.
Comment 1 - Regarding data quality, it is quite understood with figure legends and bar graphs for the efficacy of the used chemical Sodium 5-nitro guaiacol under drought stress.
Comment 2- On the other hand, this manuscript is only focused on biochemical & morphology parameters but there is no relevancy of drought stress-responsive gene functional role and expression pattern. I recommend the author characterize some of the drought stress-responsive gene expression (qPCR) or protein blot to conclude the efficacy of Sodium 5-nitro guaiacol.
Comment 3- What is the actual concentration of 5NGS used in the experiment and how you have decided on this particular dosage for treatment?
Comment 4- Add the plant figures with all the sets mentioned in the manuscript.
Comment 5- Kindly show the DAB stating the experiment to justify the level of H2O2 changes in 3d, 5d and 7d leaves.

Experimental design

Revise the material and method language content.

Validity of the findings

Data is satisfactory but no report has been given at the gene level.

Additional comments

NA

·

Basic reporting

Article meets the standards of Journal. Details enclosed in additional comments

Experimental design

Needs to mention statistical design and replications.

Validity of the findings

Agreed with authors

Additional comments

Reviewer comments on the manuscript entitled “Exogenous application of 5-NGS increased osmotic stress resistance by improving leaf photosynthetic physiology and antioxidant capacity in maize (#96044)". In this MS authors evaluated the response of exogenous spraying of 5-NGS solution on the maize seedlings leaves for alleviating adverse impact of osmotic stress. The results of study are interesting and having practical application. Although MS is fairly well written document but need refinement for grammatical and typos check. A few minor suggestions as suggested below should be considered for revision of the MS.

1. Abstract: Concise. Authors precisely mentioned the background, methodology and results with concluding remarks.
Line 30: ‘S’ should be in small case in word “Significant”
2. Introduction: Authors addressed importance of topic correctly with well identified research gap.
Line 60: Check spelling of word ‘oxyge’. It may be “oxygen”
Line 73: Mentioned the full form of “5-NGS” since abbreviation used first time.
Line 89, Mentioned the full form of “PEG” since used first time.
3. Material and Methods: Mention design and numbers of replication
Line 103, 104, 106 and whole MS: use space between numbers and unit e. g. 50 cm
Line 133: correct the formula by multiplying 100, similar to RWC formula in line 142
Line 144: ‘I’ should be small case in word “Intercellular”
4. Results: Precisely described by authors.
5. Discussion: The plant mechanisms associated with results obtained with the use of 5-NGS are well described by the authors.
6. Conclusion: concise and well written
7. Figures: Try to reduce the figure titles in short.

---

## Round 0.2 · Minor Revisions

Dear Dr Gao,

Please find the enclosed reviewers comments and request you to respond them accordingly. Kindly look into the English Language in the entire manuscript.

Reviewer 1 ·

Basic reporting

Dear Authors,
I am happy to see the implements that you have made in the manuscript according to queries. This manuscript is now well addressing the applicability of 5-NGS (Sodium 5-nitroguaiacol) for drought stress in maize

Still, I suggest adding the time point in Figure 1 legend (What is the time point that the photograph was taken after treatment- 3Days/5 Days or 7 Days as well as the age of the plant during treatment?

In the future please include molecular-level data along with biochemical and physiological parameters which might be helpful for more reliable and supportive data in your study.

I would like to recommend this manuscript for publication in Peer J.

Experimental design

Satisfactory

Validity of the findings

Satisfactory

Additional comments

I have suggested adding the time point in the Figure 1 legend for post-treatment

---

## Round 0.3 · accepted · Accept

Dear Authors,
Congratulations! your article has been accepted for publication in PeerJ.
Best wishes